# Combined Effect of Genotype, Housing System, and Calcium on Performance and Eggshell Quality of Laying Hens

**DOI:** 10.3390/ani10112120

**Published:** 2020-11-16

**Authors:** Mohamed Ketta, Eva Tůmová, Michaela Englmaierová, Darina Chodová

**Affiliations:** 1Department of Animal Husbandry, Czech University of Life Sciences Prague, 165 00 Prague–Suchdol, Czech Republic; tumova@af.czu.cz (E.T.); chodova@af.czu.cz (D.C.); 2Department of Physiology of Nutrition and Product Quality, Institute of Animal Science, 104 00 Prague–Uhříněves, Czech Republic; englmaierova.michaela@vuzv.cz

**Keywords:** genotype, littered floor, enriched cages, Ca, hen performance, eggshell

## Abstract

**Simple Summary:**

Hen performance and eggshell quality are affected by a wide range of factors from which genotype. The housing conditions and feed calcium (Ca) level might be considered the most important. Here, we compared the performance and eggshell quality of commercial hybrids (ISA Brown, Bovans Brown) and traditional Czech hybrid (Moravia BSL). Laying hens were housed in enriched cages and on littered pens and fed two different Ca levels (3.00% vs. 3.50%). Contrary to the commercial hybrids, Moravia BSL performed better under the lower feed Ca level in enriched cages. Additionally, the data pointed out the importance of studying the interaction between factors, which might help to decide the best housing system and feed Ca level for a certain hen genotype.

**Abstract:**

The objective of this study was to evaluate hen performance and eggshell quality response to genotype, housing system, and feed calcium (Ca) level. For this purpose, an experiment was conducted on 360 laying hens of ISA Brown, Bovans Brown (commercial hybrids), and Moravia BSL (traditional Czech hybrid). Laying hens were kept in enriched cages and on littered floor and fed similar feed mixtures with different Ca content (3.00% vs. 3.50%). In terms of hen performance, ISA Brown had the highest egg production (84.2%) compared to Moravia BSL (74.3%) and Bovans Brown (71.4%). Regarding eggshell quality, Bovans Brown showed the highest values of all eggshell quality parameters. Increasing feed Ca level augmented egg production (*p* ≤ 0.001) but had no effect on other performance parameters. Except eggshell thickness, all eggshell quality parameters were affected by the three-way interaction of genotype, housing, and Ca. Bovans Brown, which had the strongest eggshells (5089 g/cm^2^) when housed on a littered floor system and fed 3.00% Ca, while Moravia BSL housed on a littered floor had the weakest eggshells (4236 g/cm^2^) at 3.50% Ca. The study pointed out the importance of the interactions between studied factors on performance and eggshell quality compared to an individual factor effect.

## 1. Introduction

Eggshell quality is a trait of major economic importance related to the incidence of cracked eggs that could impair the commercial profit. The importance of the eggshell is related to its function to resist physical and pathogenic challenges from the external environment in addition to providing a source of nutrients, primarily calcium (Ca) for embryo development [1]. Therefore, maintaining laying hens’ eggshell quality is still a challenging subject for researchers.

The quality of the eggshell is affected by a wide range of factors where genotype, conditions of the housing environment, and feed Ca content might be considered the most important factors [2]. Hen performance and eggshell quality parameters vary, according to individual breeds, lines, and the genotype of laying hens. Studying hen performance differences between commercial hybrids, Singh et al. [3] reported significant differences in hen-day egg production, feed intake, and egg weight between Lohmann Brown, Lohmann White, and H&N White. Differences in eggshell strength between white and brown egg layers have been reported [4]. Moreover, different studies with brown hybrids (Isa Brown, Hisex Brown, and Moravia BSL) indicated significant differences in eggshell strength [5,6,7]. The contrast results concerning eggshell strength might be related to low heritability of eggshell strength [8]. In addition to eggshell strength, eggshell weight, eggshell thickness, and eggshell index differed between white and brown hybrids [4] and within brown hybrids [6,7].

Increasing public concern for animal welfare has pushed the poultry sector to progressively replace conventional cages for laying hens to more modern, enriched cages installed in multiple tiers within an environmentally-controlled poultry house. Perches, supplied in enriched cages, together with a nest box and scratch pads, benefit laying hen welfare with respect to egg production, health status, and behavioral repertoire [9]. In a non-cage housing system, the literature reported negative impacts on production traits including a laying percentage, egg quality parameters (egg weight, eggshell strength), and microbial contamination, which led to deleterious consequences on profitability [10,11,12]. This might be the result of excessive birds’ movement and egg-floor contact, where the floor eggs have the greatest opportunity for exposure to high levels of microorganisms. Moreover, in non-cage systems, the hen’s inappropriate use of the system resources might lead to a large proportion of eggs laid outside the nest and/or presence of droppings on a solid part of the litter and nest [13]. Englmaierová et al. [14] indicated that the highest egg production and lowest daily feed consumption were measured in cages when compared to a littered floor system. On the other hand, lower egg production and egg weight found in aviaries compared to cages was reported by Philippe et al. [13]. Ledvinka et al. [7] reported lower values of egg weight, eggshell weight, eggshell thickness, and strength in the cage housing system when compared to a littered floor system. Although the cage housing system restrict the movements of birds, it is still the ideal way to decrease the number of cracked eggs when compared to non-cage systems [15,16].

Ca is the most prevalent mineral in the bird’s body [17] and the most critical factor to ensure the proper calcification of the eggshell, since up to 96% of the dry mass of an eggshell consists of calcium carbonate [18]. Moreover, Ca plays an important role in regulating the reproductive hormones and ovary growth [19]. Ca requirements for laying hens depend on several factors such as the production phase, strain, the Ca/P ratio, and vitamin D in the diet [20]. Nutritionists and poultry producers find a challenge for determining the needs of Ca in layers but, due to the dynamic Ca requirements, the studies often resulted in conflicting outcomes. An early report by Keshavarz et al. [21] who studied the effect of three Ca levels (3.0%, 3.50% and 4.0%) showed a higher feed intake and body weight gain of the hens fed a 3.0% Ca level than for hens fed the other Ca levels with a non-significant effect on egg production and eggshell quality parameters. Jiang et al. [22] observed that layers on a diet with 2.62% Ca had a lower eggshell breaking strength than those on a diet of 3.7% or 4.4% Ca. Additionally, An et al. [23] reported a significant improvement of eggshell breaking strength and thickness when the feed Ca level increased from 3.5% to 4.7%. On the other hand, Świątkiewicz et al. [24] indicated that different Ca levels of 3.20%, 3.70%, and 4.20% did not affect the eggshell quality parameters.

Laying hen genotypes might perform differently under a certain housing system. Singh et al. [3] recommended that the hen genotype should be considered when choosing the housing system. Several studies have been done in order to evaluate the effect of a two-way interaction on hen performance and eggshell quality parameters such as the housing system and genotype [6,25], feed Ca level and age [23], genotype and feed Ca level [26], and housing system and age [27,28]. Despite that, few studies gave attention to the effect of a three-way interaction of different factors on hen performance and eggshell quality parameters.

As stated, different studies demonstrated that the performance and eggshell quality parameters of laying hens are affected by genotype, housing, and feed Ca level. Hence, a new question arises. Could hen performance and eggshell quality parameters be improved by considering the interaction of these three factors together? Therefore, the aim of this study was to compare performance and eggshell quality parameters of two commercial hybrids (ISA Brown and Bovans Brown, Hendrix genetics, Boxmeer, The Netherlands) and traditional Czech hybrid (Moravia BSL, Hendrix genetics, Boxmeer, The Netherlands) distributed between enriched cages and a littered floor system and fed two different Ca levels.

## 2. Materials and Methods

### 2.1. Ethical Approval

The experiment was approved by the Ethics Committee of the Central Commission for Animal Welfare at the Ministry of Agriculture of the Czech Republic and was carried out in accordance with Directive 2010/63/EU for animal experiments. The local Ethics Commission, case number 05/2019, approved all the procedures described in the study.

### 2.2. Experiment Design and Management

The experiment was conducted on ISA Brown, Bovans Brown (brown egg layers), and the Czech breed Moravia BSL (tinted egg layers). Moravia BSL is a three-strain hybrid of the black color bred in the Czech Republic. The egg production per housed hen is 294 eggs per year with the average egg weight of 60.7 g [29].

In total, 360 laying hens (120 hens/genotype) were used in our experiment. Sixty laying hens of each genotype were housed in 6 enriched cages SKN-O 30–60 (Kovobel, Domažlice, Czech Republic, 750 cm^2^ per hen, 60 hens, 10 hens per cage) and the other 60 hens were housed on 6 littered floor pens (6 littered pens: 60 hens, 7 hens/m^2^, 10 hens/pen). Of the 6 replicates cages and pens within the genotype, half received a diet with 3.00% Ca and half a diet with 3.50% Ca. Hence, the number of replicates per combination of genotype and diet was 3 replicates (Appendix A). All hens were beak trimmed at the same hatchery at the first day of age using an infrared trimming technique. All the conditions (including feeding regime, housing, lightning, and house temperature) were kept the same from hatching until the end of 19 weeks of age. From 20–64 weeks of age, laying hens in both housing systems were fed similar commercial feed mixtures differed in Ca content (30 g/kg feed vs. 35 g/kg feed), where the source of the additional Ca was limestone grains. The complete composition of the feed mixtures is given in Table 1. The littered floor pens were covered by wood shavings. The floor area had a 40 cm feeder, two nipple water dispensers, and two nests (30 cm × 30 cm) delimited by plastic curtains. Litter on the floor was not removed until the end of the production. Enriched cages were conforming to the Council Directive 1999/74/EC. They were equipped with two steel perches (75 cm) and two nests (30 cm × 30 cm) delimited by plastic curtains. A pecking and scratching area were located above the nests. A linear front feeder provided 120 cm access to the hens. In each cage, there were three nipple drinkers. Underneath the cage was a manure belt conveyor for manure removal, which was removed twice a week. Feed and drinking water were available on an *ad libitum* basis. The light schedule was identical in both systems where the daily photoperiod consisted of 16L:8D. The lights were turned on at 3:00 a.m. and off at 07:00 p.m.

### 2.3. Analytical Determination

The chemical composition analysis of feedstuff was determined according to the Association of Official Analytical Chemists (AOAC) methods [30] for crude protein (index no. 954.01) and ash contents (index no. 942.05). Gross energy was determined by combustion with an adiabatic bomb calorimeter (Laget MS10A, Laget, Prague, Czech Republic). Ca content was analyzed using the AOAC method (index no. 965.17), based on a vanad-molybden reagent and spectrophotometry analysis on Solaar M6 apparatus (TJA Solutions, Cambridge, UK).

### 2.4. Hen Performance

During the experiment, egg production was recorded daily. The data of egg production were used to calculate hen-day egg production as the number of eggs produced during a period of time divided by the number of days in that period and multiplied by 100. Feed intake was recorded daily per pen or cage and calculated as an average for a hen. Based on feed intake, daily Ca intake was calculated. Egg weight (2421 eggs in total per genotype) was determined with an electronic balance (Metler Toledo Inc., Columbus, OH, USA, resolution of 0.01 g).

### 2.5. Eggshell Physical Properties

Eggs for eggshell quality assessments were collected every four weeks on two consecutive days at 9:00 in the morning to be 2421 eggs per genotype in total. Measurements of eggshell quality were done at the same day of collection. The lengths and widths of each freshly laid egg were measured for the egg shape index calculation (width × length^−1^ × 100). Eggshell strength was determined using shell strength and packaging analyser (QC-SPA) device (TSS, York, England, UK). After measuring eggshell strength, the eggs were broken, and the internal egg components were discarded. Eggshell thickness was measured with a QCT shell thickness micrometer (TSS, York, England, UK) at the equatorial area after removal of shell membranes. Eggshell weight was determined after complete drying. The eggshell index was calculated as eggshell weight of an egg divided by eggshell surface of the egg and multiplied by 100, where the surface area of each egg was calculated using the formula: egg surface = 4.67 egg weight^2/3^ [31].

### 2.6. Statistical Analyses

The data were statistically evaluated using the General Linear Models procedure in SAS software [32]. All the reported data are mean values that were tested for normality with the Shapiro-Wilk test. The data were analyzed by a three-way ANOVA with interactions between genotype, housing system, and feed Ca level. Statistically significant differences were assessed using Tukey’s adjustment test. All differences were considered significant at *p* ≤ 0.05.

## 3. Results

### 3.1. Hen Performance

The individual effect of the genotype, housing system, and feed Ca levels as well as the interaction between these factors on hen performance parameters and egg weight are presented in Table 2. Hen genotype significantly affected the hen-day egg production (*p* ≤ 0.001). ISA Brown had the highest egg production (84.2%) compared to Moravia BSL (74.3%) and Bovans Brown (71.4%). Significantly higher hen-day egg production (*p* ≤ 0.05) was found in enriched cages (80.2%) than on a littered floor system (74.6%). Regarding the Ca effect, the highest hen-day egg production (*p* ≤ 0.05) was recorded at the 3.50% Ca level (79.6%) while the lowest production (73.7%) was at the 3.00% Ca. The hen-day egg production was significantly affected by the three-way interaction of genotype, housing, and feed Ca level (*p* ≤ 0.001). Hen-day egg production of ISA Brown housed in enriched cages at 3.50% Ca was significantly the highest (89.6%) and the lowest (55.8%) was in Bovans Brown at 3.00% Ca in the same housing.

Moravia BSL consumed more daily feed (*p* ≤ 0.01) and daily Ca (*p* ≤ 0.01) than Bovans Brown and ISA Brown. Littered floor system showed significantly higher (*p* ≤ 0.001) daily feed intake (193 g) and daily Ca intake (6.32 g) compared to enriched cages (125 g and 4.13 g, respectively). Our study indicated a non-significant effect of feed Ca level on daily feed intake and daily Ca intake. However, we found slightly higher daily feed intake and daily Ca intake as the feed Ca level decreased. Feed intake and Ca intake were significantly the highest (*p* ≤ 0.01) in Moravia BSL hens housed on a littered floor system at 3.00% Ca (217 g and 8.04 g, respectively) while ISA Brown housed in enriched cages fed on both levels of Ca showed identically lower values (122 g and 3.98 g, respectively). The heaviest eggs (*p* ≤ 0.001) were produced by Bovans Brown (63.2 g) while ISA Brown had the lightest eggs (61.5 g). Eggs laid on a littered floor system had significantly higher egg weights (*p* ≤ 0.001) compared to enriched cages. A non-significant effect on feed Ca level on egg weight was detected in our study. Egg weight was affected by the three-way interaction between studied factors (*p* ≤ 0.001). The heaviest eggs (64.3 g) were produced by Bovans Brown on a littered floor system with a 3.00% feed Ca level, while ISA Brown housed in enriched cages and fed 3.50% Ca had the lightest eggs (59.2 g).

### 3.2. Eggshell Quality Parameters

The individual effect of genotype, housing system, and feed Ca level as well as the interaction between these factors on eggshell quality parameters are shown in Table 3. A significant effect of hen genotype was recorded for eggshell weight (*p* ≤ 0.001). Eggs laid by Bovans Brown had heavier eggshells (6.5 g) compared to ISA Brown (6.1 g) and Moravia BSL (5.8 g). A non-significant effect of the housing system on eggshell weight was detected in this study. Increasing feed Ca level from 3.00% to 3.50% significantly decreased the eggshell weight (*p* ≤ 0.05) (from 6.2 g to 6.0 g). The three-way interaction between evaluated factors was obtained for eggshell quality parameters in our study. Bovans Brown hens housed on a littered floor system and in enriched cages at 3.00% Ca had identically the highest eggshell weight (6.7 g) while the lightest eggshells (5.6 g) were found in eggs laid by Moravia BSL hens housed in enriched cages at a 3.00% Ca.

Bovans Brown eggs showed the highest values (*p* ≤ 0.001) of eggshell strength (4839 g/cm^2^) compared to Moravia BSL (4369 g/cm^2^) and ISA Brown (4265 g/cm^2^). Differences in eggshell strength between the littered floor system and enriched cages were not significant in our study. However, enriched cages showed slightly stronger eggshells (4654 g/cm^2^) compared to a littered floor system (4604 g/cm^2^). The eggshell strength decreased (from 4673 g/cm^2^ to 4590 g/cm^2^) as the feed Ca level increased (*p* ≤ 0.01). Regarding the three-way interaction, the strongest eggshells (*p* ≤ 0.001) were produced on a littered floor system by Bovans Brown fed 3.00% Ca (5089 g/cm^2^) while Moravia BSL hens housed on a littered floor had the weakest eggshells (4236 g/cm^2^) at 3.50% Ca. Bovans Brown produced eggs of thicker eggshells (0.373 mm) in comparison with ISA Brown (0.362 mm) and Moravia BSL eggshells (0.330 mm). Non-significant interaction between evaluated factors was detected for eggshell thickness. The highest egg shape index values were observed for eggs laid by Bovans Brown (77.5%) while eggs laid by Moravia BSL showed the lowest egg shape index values (77.0%). The three-way interaction between evaluated factors significantly affected the egg shape index (*p* ≤ 0.001) where the highest values were found in eggs of Bovans Brown eggs housed on a littered floor system and fed 3.00% Ca. Eggshell index values of Bovans Brown eggs that were significantly the highest (*p* ≤ 0.001). When the feed Ca level increased from 3.00% to 3.50%, the eggshell index significantly decreased from 10.1 g/100 cm^2^ to 9.9 g/100 cm^2^. Regarding the three-way interaction, eggshell index values were identically the highest (10.7 g/100 cm^2^) in eggs laid by Bovans Brown eggs housed on both housing systems and fed 3.00% Ca, while Moravia BSL hens housed in enriched cages on a 3.00% Ca level had the lowest values (9.5 g/100 cm^2^).

## 4. Discussion

In terms of productivity and feed intake, ISA Brown hens had the best performance in our study with the highest egg production and the lowest feed and Ca intake. Our results also indicated differences in egg weight between studied genotypes where Bovans Brown hens produced the heaviest eggs while ISA Brown hens had the lightest eggs. Similar findings were observed by Singh et al. [3] and Tůmová et al. [33] who reported a difference in egg weight according to different hen genotypes. The data of our study showed that ISA Brown and Moravia BSL laid eggs of lower egg weight (62.7 g and 63.5 g, respectively) compared to breed standard statements (63.1 g and 64.5 g, respectively), while Bovans Brown laid heavier eggs (64.3 g) when compared to breed standard statements (63.3 g) [29,34,35]. The differences in egg weight reported in our study might be related to the higher productivity of the ISA Brown genotype (84.2%) compared to Moravia BSL (74.3%) and Bovans Brown (71.4%). Moravia BSL showed significantly higher daily feed intake compared to ISA Brown and Bovans Brown. Poor plumage conditions (observed but not recorded) could be the reason for the higher feed intake by Moravia BSL. Moravia BSL had one dead hen in the last week of the experiment. Otherwise, no mortality was observed for the other used genotypes during the whole experimental period. Therefore, mortality was not included in our study.

Hens housed in enriched cages had significantly higher hen-day egg production when compared to a littered floor system. Our results agree with Anderson et al. [36] who reported higher egg production from the cage housing system. The experiment of Englmaierová et al. [14] showed that the laying rate was about 80% with a littered floor system compared to about 92% with both conventional and enriched cages. Littered floor system showed higher daily feed intake and daily Ca intake when compared to enriched cages, which is in correspondence with the study of Tůmová et al. [37]. The authors reported a higher feed intake from a non-cage system. On the other hand, lower feed intake in a non-cage housing system was reported by Karcher et al. [11]. In our study, the higher daily feed intake observed on the littered floor system could be explained by losses of feed caused by high birds’ movements.

Egg weight is an important trait that influences egg quality and egg grading where it is determined without breaking the egg [27]. Eggs produced on a littered floor system were significantly heavier than those from enriched cages in our study. Different results were reported by Englmaierová et al. [14] who found heavier eggs in enriched cages compared to a littered floor system. Samiullah et al. [27] reported that the egg weight was reduced by 2 g with a non-cage system compared to conventional cages (58.6 versus 60.7 g, respectively). The variation among reports regarding the housing system effect on hen performance might be caused by a multiple factors’ effect such as environmental conditions and age.

Our results indicated an increase in hen-day egg production since the feed Ca level increased from 3.00% to 3.50%. Consistent results were reported by Safaa et al. [26] who obtained an improvement of egg production with an increasing Ca level. On the other hand, An et al. [23] and Tůmová et al. [37] reported a non-significant effect of feed Ca level on egg production. The variation among reports on the effect of Ca on egg production could be due to multiple factors interfering such as genotype and feeding regime. Our study also indicated a non-significant effect of the feed Ca level on daily feed intake, daily Ca intake, and egg weight, which is in correspondence with An et al. [23] and Cufadar et al. [38]. Narvaez-Solarte et al. [39] reported that daily feed intake was decreased as dietary Ca level increased. This observation is presumably related to the over consumption of feed to restore the lack of Ca.

The three-way interaction of genotype, housing, and feed Ca level appeared to play an important role on hen performance since all the parameters were significantly affected. Hen-day egg production of ISA Brown hens housed in enriched cages at the 3.50% Ca level was significantly the highest (89.6%) and the lowest was in Bovans Brown at 3.00% Ca in the same housing. These findings differ with our previous study [37], where non-significant interaction between hen genotype, housing, and feed Ca level were detected for hen-day egg production. The conflict results are presumably due to different laying hen genotypes used in the previous experiment (Lohmann LSL and Czech Hen). However, Bovans Brown housed in both systems and fed 3.00% Ca had the lowest hen-day egg production of all evaluated genotypes. It might be assumed that, for Bovans Brown, the Ca level of 3.00% was not sufficient for egg production.

Eggshell characteristics are important measures of quality, determining hatchability and preference by consumers for table eggs. In our study, Bovans Brown genotypes showed the best eggshell quality parameters in comparison with ISA Brown and Moravia BSL genotypes. Similarly, better eggshell quality values of eggs produced by Bovans Brown genotype compared to other brown studied genotypes [33]. Franco-Jimenez and Beck [40] reasoned the better eggshell quality of brown eggs to the greater bone frame and stronger bone structure of brown egg layers that allow them to store a greater amount of Ca, leading to better eggshell formation. Additionally, Yang et al. [41] reported a high correlation between dark eggshell color and eggshell quality. We might assume that the better eggshell parameters of Bovans Brown eggs are related to the higher egg shape index as the higher egg shape index values the more force needed to rapture the egg [42]. Moreover, a positive correlation between the egg shape index and eggshell strength has been reported [43], where the egg shape index might be an indicator for overall eggshell quality parameters. Results of eggshell strength showed stronger eggshells of ISA Brown and Bovans Brown eggs (4773 g/cm^2^ and 5089 g/cm^2^, respectively) compared to breed standard statements (4100 g/cm^2^ and 4050 g/cm^2^, respectively) [34,35].

There is a large degree of variability in the research findings on the effect of housing system on egg weight and eggshell quality parameters providing unclear indication of which production system maintains eggs with the best eggshell quality [14]. A non-significant effect of the housing system was detected in our study for eggshell quality parameters. A non-significant effect of the housing system on some eggshell quality parameters, namely eggshell strength and thickness, were detected in our previous study [25]. Moreover, Kühn et al. [44] concluded that the housing system had no effect on the eggshell weight and eggshell thickness of eggs from the littered floor and free-range systems.

Increasing the feed Ca level significantly decreased the eggshell weight, eggshell strength, and eggshell index. Contrary findings were indicated by Yang et al. [41] and Świątkiewicz et al. [24] who reported a non-significant effect of the Ca level on eggshell quality parameters. In our study, the higher eggshell quality parameters detected at a lower feed Ca level might be reasoned to the slightly higher daily feed intake and daily Ca intake (172 g and 5.64 g, respectively) at the 3.00% Ca level compared to 166 g and 5.46 g, respectively, at a 3.50% Ca level. It can be assumed that this extra feed was utilized in eggshell formation.

Our study showed a significant interaction of housing system, hen genotype, and feed Ca level on the majority of eggshell quality parameters. Eggs of heavier eggshells were laid by Bovans Brown on a littered floor system and in enriched cages at a lower feed Ca level. Similar trends were reported by Tůmová et al. [37] who observed significant interactions between genotype, housing, and feed Ca level for eggshell weight. Bovans Brown housed on the littered floor system and fed the lower level of Ca showed the highest values of eggshell strength, egg shape index, and eggshell index. The higher eggshell quality parameters of Bovans Brown hens could be related to lower egg production detected in our study. It might be explained that the interaction between factors could affect the final product. For instance, the commercial hybrids are selected for highly controlled conditions and might not be suitable for non-cage systems, which correspond more to traditional breeds.

The data of our study showed that ISA Brown, Bovans Brown, and Moravia BSL laid eggs of lower egg weight (61.5 g, 63.2 g, and 62.6 g, respectively) compared to breed standard statements (63.1 g, 63.6 g, and 64.5 g, respectively). On the other hand, results of eggshell strength showed stronger eggshells of ISA Brown and Bovans Brown eggs (4265 g/cm^2^ and 4839 g/cm^2^, respectively) compared to breed standard statements (4100 g/cm^2^ and 4050 g/cm^2^, respectively).

## 5. Conclusions

In conclusion, the data showed that the hen genotype had a great effect on hen performance and eggshell quality parameters even within brown egg layers. Regarding hen performance parameters, the housing system affected the results more than the Ca level, whereas, in terms of eggshell quality, a higher impact of Ca level was observed. The interaction of all evaluated factors was more important in eggshell quality than in hen performance parameters. Contrary to the commercial hybrids, Moravia BSL performed better under a lower feed Ca level in enriched cages. Bovans Brown fed a Ca level of 3.00% laid eggs of higher eggshell qualities. We can assume that, in this genotype, layers better utilized Ca, which led to better eggshell quality parameters.

## Figures and Tables

**Table 1 animals-10-02120-t001:** Composition of experimental diets and nutrient content on a dry matter basis.

Item	Content (%)	
	Feed Mixture 3.0% Ca	Feed Mixture 3.5% Ca
Wheat	35.8	35.5
Maize	31.0	33.3
Soya extracted meal	15.5	15.5
Fish meal	1.5	1.5
Wheat bran	2.5	2.5
Alfalfa meal	3.0	2.0
Rapeseed oil	3.0	3.0
Limestone	6.8	8.0
Dicalcium phosphate	1.0	1.0
Sodium chloride	0.2	0.2
Vitamin-mineral premix ^a^	0.5	0.5
Analysed Content of Nutrients		
Crude protein (g/kg)	155.2	153.7
AME (MJ/kg)	11.54	11.58
Calcium (%)	3.03	3.48
Phosphorus (%)	0.56	0.56

^a^ Vitamin-mineral premix provided per kg of diet: retinyl acetate 8000 IU (international unit), vitamin D3 2250 IU, vitamin E 15 mg, menadione 1.5 mg, thiamine 1.5 mg, riboflavin 4 mg, pyridoxine 2 mg, cobalamin 0.01 mg, niacinamide 20 mg, Ca pantothenate 6 mg, biotin 0.06 mg, folic acid 0.4 mg, choline chloride 250 mg, betaine 50 mg, DL-methionine 0.3 g, Co 0.3 mg, Cu 6 mg, Fe 30 mg, I 0.7 mg, Mn 60 mg, Zn 50 mg, Se 0.2 mg. AME: apparent metabolizable energy.

**Table 2 animals-10-02120-t002:** The effects of genotype, housing system, and feed Ca level and their interaction on hen performance parameters and egg weight.

Factor	Item		Hen-Day Egg Production (%)	Daily Feed Intake (g)	Daily Ca Intake (g)	Egg Weight (g)
**Effect of Individual Factors**						
Genotype RMSE	ISA Brown		84.2 ^a^	146 ^c^	4.72 ^c^	61.5 ^c^
	Bovans Brown		71.4 ^c^	175 ^b^	5.71 ^b^	63.2 ^a^
	Moravia BSL		74.3 ^b^	186 ^a^	6.01 ^a^	62.6 ^b^
			20.3	67.5	2.19	6.10
Housing RMSE	Littered floor		74.6 ^b^	193 ^a^	6.32 ^a^	62.9 ^a^
	Enriched cages		80.2 ^a^	125 ^b^	4.13 ^b^	61.7 ^b^
			20.8	61.3	1.99	6.12
Ca RMSE	3.00		73.7 ^b^	172	5.64	62.1
	3.50		79.6 ^a^	166	5.46	62.1
			20.8	69.3	2.25	6.14
*p*-value	Genotype		***	**	**	***
	Housing		*	***	***	***
	Ca		*	NS	NS	NS
**Analysis of Interaction between the Factors**						
ISA Brown	Littered floor	3.00	78.4 ^b^	157 ^bc^	5.11 ^bc^	62.7 ^bc^
		3.50	85.8 ^ab^	162 ^bc^	5.26 ^bc^	62.5 ^bc^
	Enriched cages	3.00	86.1 ^ab^	122 ^d^	3.98 ^d^	60.4 ^cd^
		3.50	89.6 ^a^	122 ^d^	3.98 ^d^	59.2 ^d^
Bovans Brown	Littered floor	3.00	65.9 ^c^	191 ^b^	6.21 ^ab^	64.3 ^a^
		3.50	80.3 ^b^	210 ^ab^	6.83 ^ab^	61.9 ^c^
	Enriched cages	3.00	55.8 ^d^	126 ^cd^	4.11 ^b^	63.3 ^ab^
		3.50	80.2 ^b^	129 ^cd^	4.22 ^b^	63.4 ^ab^
Moravia BSL	Littered floor	3.00	64.9 ^c^	217 ^a^	8.04 ^a^	62.8 ^ab^
		3.50	72.4 ^bc^	191 ^b^	6.21 ^ab^	63.5 ^ab^
	Enriched Cages	3.00	86.8 ^ab^	126 ^cd^	4.12 ^cd^	61.3 ^cd^
		3.50	82.3 ^ab^	124 ^cd^	4.05 ^cd^	62.4 ^bc^
RMSE			19.2	57.7	1.88	6.02
*p*-value Genotype × Housing			*	*	*	***
Genotype × Ca			**	NS	NS	NS
Housing × Ca			NS	NS	NS	NS
Genotype × Housing × Ca			***	**	**	***

Results of the variance analysis are indicated as significant (* *p* ≤ 0.05, ** *p* ≤ 0.01, *** *p* ≤ 0.001) or not significant (NS). ^a,b,c,d^ Mean values in the same column marked with a different superscript indicate statistical significance (*p* ≤ 0.05). Mean values with no superscript are not significantly different from any other values. RMSE: Root Mean Square Error.

**Table 3 animals-10-02120-t003:** The effects of genotype, housing system, and feed Ca level and their interactions on eggshell quality parameters.

Factor	Item		Shell Weight (g)	Shell Strength (g/cm^2^)	Shell Thickness (mm)	Egg Shape Index (%)	Shell Index (g/100 cm^2^)
**Effect of Individual Factors**							
Genotype RMSE	ISA Brown		6.1 ^b^	4265 ^c^	0.362 ^b^	77.2 ^b^	9.9 ^b^
	Bovans Brown		6.5 ^a^	4839 ^a^	0.373 ^a^	77.5 ^a^	10.5 ^a^
	Moravia BSL		5.8 ^c^	4369 ^b^	0.330 ^c^	77.0 ^c^	9.6 ^c^
			0.69	848	0.18	2.80	0.74
Housing RMSE	Littered floor		6.2	4604	0.353	77.2	10.0
	Enriched cages		6.1	4654	0.363	77.3	10.0
			0.74	865	0.187	2.81	0.81
Ca RMSE	3.00		6.2 ^a^	4673 ^a^	0.365	77.3	10.1 ^a^
	3.50		6.0 ^b^	4590 ^b^	0.352	77.2	9.9 ^b^
			0.74	864	0.187	2.81	0.81
*p*-value	Genotype		***	***	**	**	***
	Housing		NS	NS	NS	NS	NS
	Ca		*	**	NS	NS	*
**Analysis of Interaction between the Factors**							
ISA Brown	Littered floor	3.00	6.1 ^b^	4408 ^cd^	0.348	77.1 ^ab^	9.8 ^ab^
		3.50	6.3 ^ab^	4773 ^ab^	0.361	76.6 ^b^	10.1 ^ab^
	Enriched cages	3.00	5.9 ^bc^	4723 ^ab^	0.391	77.8 ^ab^	9.9 ^ab^
		3.50	5.8 ^bc^	4638 ^ab^	0.352	77.4 ^ab^	9.9 ^ab^
Bovans Brown	Littered floor	3.00	6.7 ^a^	5089 ^a^	0.381	78.1 ^a^	10.7 ^a^
		3.50	6.3 ^ab^	4868 ^ab^	0.368	77.9 ^ab^	10.3 ^ab^
	Enriched cages	3.00	6.7 ^a^	4943 ^a^	0.383	77.1 ^ab^	10.7 ^a^
		3.50	6.4 ^ab^	4562 ^bc^	0.363	77.3 ^ab^	10.3 ^ab^
Moravia BSL	Littered floor	3.00	5.8 ^bc^	4328 ^cd^	0.327	76.6 ^b^	9.6 ^b^
		3.50	5.8 ^bc^	4236 ^d^	0.328	77.3 ^ab^	9.6 ^b^
	Enriched Cages	3.00	5.6 ^c^	4599 ^bc^	0.331	77.5 ^ab^	9.5 ^b^
		3.50	5.8 ^bc^	4384 ^cd^	0.333	76.8 ^b^	9.7 ^ab^
		RMSE	0.68	835	0.186	2.79	0.73
*p*-value Genotype × Housing			***	***	NS	***	NS
Genotype × Ca			***	***	NS	*	***
Housing × Ca			***	***	*	NS	***
Genotype × Housing × Ca			***	***	NS	***	***

Results of the variance analysis are indicated as significant (* *p* ≤ 0.05, ** *p* ≤ 0.01, *** *p* ≤ 0.001) or not significant (NS). ^a,b,c,d^ Mean values in the same column marked with a different superscript indicate statistical significance (*p* ≤ 0.05). Mean values with no superscript are not significantly different from any other values. RMSE: Root Mean Square Error.

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
