# Peer review of "Combined Effect of Genotype, Housing System, and Calcium on Performance and Eggshell Quality of Laying Hens"

_animals, 2020, doi:10.3390/ani10112120_

Round 1
Reviewer 1 Report
I am very satisfied with the revisions made to the manuscript by the Authors. Apart from few corrections and clarifications that need to be made, the paper is ready for publication. Please refer to the specific comments below.
Figure 1: Although I am very satisfied with the authors’ attempt to include a figure describing the experimental design, the figure still needs a minor rework. Firstly, the figure’s caption should be placed below the figure, not above, with the “Figure 1” in bold. Please present the figure as a picture, not as a graph (replace the color bars), as the “1, 2, 3, 4, 5, 6, 7, 8, 9, 10” to the left, with no description, is confusing. Also, please include more descriptions in the figure, e.g. concerning the number of replicates (n=3), the number of hens per cage (“10”) or per genotype (“120 hens” instead of just “120”). As a reference, please see another paper in Animals - https://www.mdpi.com/2076-2615/10/9/1539. If I may advise, PowerPoint is a very suitable tool for preparing such simple figures.
Line 119: If the “conditions were kept the same” (lines 114-115), please replace “similar” with “identical”.
Line 116 and Table 1: The table format has changed and now does not match the Animals template (that goes for all of the tables). Please add the previously removed “(%)” in a following way, as the “%” refers to the values presented in the table, not to the listed items:
|
Item |
Content (%) |
|
|
Feed mixture 3.0% Ca |
Feed mixture 3.5% Ca |
|
Moreover, my comment referring to the line 81 in the previous version of the manuscript (line 116 in the revised version) was not addressed. Based on the data from the table the, feed mixtures are not identical. Given the experimental design, I assume there was one batch of the initial, basal feed separately amended and mixed with limestone for each treatment group. Thus, the only difference between treatment feeds groups should be in the limestone content, but there are differences in maize (31.0 vs 33.3), wheat (35.8 vs 35.5) and alfalfa meal (3.0 vs 2.0). As I understand that the values presented in the table are given for the treatment feeds after they were prepared and then analyzed for the content (thus the differences in % of the ingredients) the values in the table does not match the description in the text. So either the values in the tables need to be revised or the description of the treatment feeds preparation needs to be amended, because the feeds are not identical, but rather “similar”.
Line 132 and citation [25]: Association of Official Analytical Chemists (please add the missing “s”).
Line 154: Consider removing the brackets and changing to: “… using the formula: egg surface = 4.67 egg weight2/3.” for better readability.
Statistical Analysis: Thank you for the explanation. However, in any of the provided papers there is no information if the data were tested for normality. It is good that the data were normally distributed in this experiment, but nonetheless it is very important to include that information in this section, including what statistical test has been used to evaluate the normality. I will stress again that, to perform the analysis of variance (ANOVA), the variable should be approximately normally distributed. Otherwise ANOVA analysis is unreliable.
Line 215: Please provide the value for the lightest eggshells.
Lines 303-304: Expression “suitability for table” is unclear. Please reword.
Line 332: Please reword “obtained” to “showed”.
Line 344: Consider changing “egg type of hens” to “layers” or “laying hens”.
Lines 348-351: “In Bovans Brown, Ca level of 3.00% was not sufficient for egg production and birds were not able, at this low Ca level, to compensate by feed and Ca intake. We can assume that in this genotype layers better utilized Ca, which led to better eggshell quality parameters.” Please reword both sentences as they seem right now to stay in contradiction: “was not sufficient” vs. “better utilized, … better … parameters”.
References: “Doi” should be written as “DOI” (capital letters).
Best regards and good luck,
Reviewer
Author Response
Dear reviewer,
We would like to thank you for evaluating our manuscript.
Please see the attachment.
Kind regards

Reviewer 2 Report
Thank you to the authors for considering the proposed changes, and the manuscript is much improved. The authors have addressed most of my comments. The manuscript would benefit from some English language improvement.
I would suggest that the authors look up the Coalition for Sustainable Egg Supply. There is more recent literature from that research project that are directly related to the manuscript. All papers were published in Poultry Science.
To clarify the previous comment on the Scheffe's test. I included a typo, and meant post hoc, not post doc test. I am not sure why this was deleted completed, that was not my intent. The post hoc test used should be stated, however, as their are many choices of post hoc tests, it is good to cite a reference that helped you make the choice.
The response to the comment on the hens age is not satisfactory. I strongly think age should be included in the analysis, as this could have changed differentially and would be important for the current investigation. This is a very large time period in the life cycle of the hens, where we know differences can occur.
The response to the comment on comparing genotypes to their standards is also not satisfactory. Knowing how they produced under the current treatments compared to what the standard production is puts the current results into context of the standard production. Did they produce better, worse, the same than what the breed standard states?
Author Response

(The authors gave the same response as above.)

Reviewer 3 Report
The revised version responds to all the observations I made: so, I find now the paper can be accepted.Author Response
Dear reviewer,
We would like to thank you for evaluating our manuscript.
Please see the attachment.
Kind regards

Reviewer 4 Report
General comments
The calcium-metabolism is highly dependent on the relation between Ca and P, and also vitamin D, and I am surprised that this is not at all mentioned. It is not clear what level of Ca that is normal (is 3.5 to be considered high and 3.0 normal etc) and why was these inclusion levels chosen.
The Introduction needs a general improvement. It is very detailed regarding specific comparisons between genotypes and housing systems in earlier studies, this can be shortened. Instead use some lines to introduce the reason for choosing these Ca-levels.
Line 47 – it is mentioned that a high rate of shell damage occurs during egg collection, sorting and transportation. Yes, if equipment is of poor design, but I do not think this is a major problem. But nothing is mentioned about reasons to shell damage in the housing system.
Line 54 – not clear in what way a better well-being automatically leads to higher quality products.
M &M
Line 105 -107 – detailed info about Moravia BSL. M & M is not the right place for this detailed information of expected performance.
Figure 1 is not needed. You can easily explain the experimental set up in the text.
I think that 3 replicates is a short-coming. I think that 4 is a minimum for this kind of study.
Line 138 - “Data of egg production were used to calculate hen-day egg production as the number of eggs produced during a period of time divided by the number of days in that period and multiplied by 100.” What was the period used ( was it the whole length of the study)? You must also divide by the number of layers.
Housing system is one of the factors focused on. Surprisingly, the housing systems are not well described. Besides stocking density nothing at all is mentioned regarding the littered pen. Was the pen enriched with perches, nests? Removal of manure? Feeding equipment etc. I think it is difficult to draw general conclusions regarding differences between enriched cages and floor housing when such small groups are applied on floor.
Statistical analyses
The data was analyzed with GLM in SAS with ANOVA. I lack information about whether analyses was performed on replicate means, i.e., was one average calculated for each replicate before analyzing performance parameters in table 2? Regarding egg shell physical properties – these were recorded on eggs every 4 week during the study. How was age treated in the analyses? Was age included in the model and in that case, was adjustment for repeated measurements performed?
As multiple comparisons are performed for both performance data and egg quality some kind of adjustment for multiple comparisons should be made, e.g. Tukey adjustment.
Results
I am wondering about the extremely high feed intake. I assume that feed spillage occurred in the floor pens, which is mentioned in the Discussion. But is it then relevant to calculate a daily Ca intake, when figures likely are biased by feed waste? Were the birds beak trimmed? How was the feather coverage? A poor plumage condition will increase feed intake.
Mortality is not mentioned at all, although I think it is relevant.
Discussion
Line 275 – not clear which behavioural needs that layers in the littered floor system could fulfil since the environment has not been described
Line 269 – in what way does higher animal activity and competition for facilities lead to reduced egg production? Feed intake was extremely high, so a feed restriction due to competition is not likely.
I would like a reflection regarding how differences in feed intake between floor housed birds and birds in cages can lead differences in production performance. If we assume that the difference in feed intake is not only due to feed waste. E.g., did birds differ in body weight?
Caution should be taken when comparing the results from this study with results from studies using floor pens with larger groups of layers. 10 birds is more to consider as an experimental environment rather than a floor housing system.
Author Response

(The authors gave the same response as above.)

Round 2
Reviewer 4 Report
The paper has been improved, but I would still like to see a further improvement regarding e.g. statistics. I do not understand why age is not included in the statistical model. Please, see my comments in the pdf file.

Author Response
Dear reviewer,
Thank you for your evaluation.
Please see the attachment.
Kind regards

This manuscript is a resubmission of an earlier submission. The following is a list of the peer review reports and author responses from that submission.
Round 1
Reviewer 1 Report
Dear Authors, Please refer to the attached .docx file.

Author Response
Dear reviewer,
Please see the attachement

Reviewer 2 Report
Introduction:
Overall, the introduction does not help the reader understand why the current experiment was conducted. A hypothesis or prediction should be included in the last paragraph along with the objective. There is also more recent literature that can be incorporated into the introduction, particularly on housing systems.
P1L22-23: There appears to be an extra feed in this sentence.
P2L49: Change form to from.
Methods:
There should be some description of the nutrition for the hens prior to the study (20 wks). Was the diet the same, were they housed the same? Presumably these early effects could influence the results.
Why were internal egg quality measures not taken?
Why did the authors choose a Scheffe's test as a post doc? This should be justified with a citation.
Did the authors take age of the hen into account? Presumably the number of eggs laid per day changed over the 20 to 64 weeks of age. Was the change similar?
Results and discussion:
Table 2 is difficult to read. Please ensure all columns have headers.
Please include test statistics (df at a minimum).
P5L146-147: It is very speculative to state that social stress and movement were causes of observed differences when these were not measured.
It is important here to have some comparison of the genotypes to their standards. Did they perform as suggested by the breeder, better, worse?
Author Response

(The authors gave the same response as above.)

Reviewer 3 Report
This research tried to demonstrate the interaction of genotype, housing conditions, and feed calcium (Ca) levels effects on performance and eggshell quality of Laying Hens. However, the design is too complicated, and the statistic analysis was not proper in this study. The comments for this manuscript were labeled on the pdf file, please check.

Author Response

(The authors gave the same response as above.)

Reviewer 4 Report
Why did you decided that the experiment lasted at 20 to 64 weeks?
Why you did not predict the effects of hen age on egg quality?
Genotype have effect on egg quality, but age of laying hens also, it affects the number of eggs, weight and quality of the shell.
Since you did not include this effects in your research, it is possible to assume that the age of hens had an impact on some quality indicators.
One should be very careful when concluding.
Author Response

(The authors gave the same response as above.)
